# Interaction with the Assembly Chaperone Ump1 Promotes Incorporation of the β7 Subunit into Half-Proteasome Precursor Complexes Driving Their Dimerization

**DOI:** 10.3390/biom12020253

**Published:** 2022-02-04

**Authors:** Jessica Zimmermann, Paula C. Ramos, R. Jürgen Dohmen

**Affiliations:** Institute for Genetics, Center of Molecular Biosciences, Department of Biology, Faculty of Mathematics and Natural Sciences, University of Cologne, 50674 Cologne, Germany; jzimme11@smail.uni-koeln.de (J.Z.); pramos@uni-koeln.de (P.C.R.)

**Keywords:** proteasome, assembly, *Saccharomyces*, 15S precursor complex, Ump1, Pre4, Rpn4

## Abstract

Biogenesis of the eukaryotic 20S proteasome core particle (PC) is a complex process assisted by specific chaperones absent from the active complex. The first identified chaperone, Ump1, was found in a precursor complex (PC) called 15S PC. Yeast cells lacking Ump1 display strong defects in the autocatalytic processing of β subunits, and consequently have lower proteolytic activity. Here, we dissect an important interaction of Ump1 with the β7 subunit that is critical for proteasome biogenesis. Functional domains of Ump1 and the interacting proteasome subunit β7 were mapped, and the functional consequences of their deletion or mutation were analyzed. Cells in which the first sixteen Ump1 residues were deleted display growth phenotypes similar to *ump1*∆, but massively accumulate 15S PC and distinct proteasome intermediate complexes containing the truncated protein. The viability of these cells depends on the transcription factor Rpn4. Remarkably, β7 subunit overexpression re-established viability in the absence of Rpn4. We show that an N-terminal domain of Ump1 and the propeptide of β7 promote direct interaction of the two polypeptides in vitro. This interaction is of critical importance for the recruitment of β7 precursor during proteasome assembly, a step that drives dimerization of 15S PCs and the formation of 20S CPs.

## 1. Introduction

Intracellular protein degradation is essential for protein quality control, as well as the regulation of cellular processes. A central player in cellular proteolysis is the 26S proteasome, which is composed of two regulatory particles attached to the distal ends of a catalytic core particle (CP), the 20S proteasome [1,2,3,4]. Biogenesis of the CP is a process assisted by specific chaperones that are absent from the active complex. In *Saccharomyces cerevisiae*, these are the two dimeric chaperones Pba1-Pba2 and Pba3-Pba4, as well as Ump1, the first proteasome-specific chaperone identified [5]. In contrast to the two dimeric Pba chaperones that are recycled during the biogenesis process, Ump1 is degraded by the native 20S proteasome upon completion of its assembly [5,6]. Ump1 was first detected in a precursor complex (PC) called 15S PC, a half CP with one copy of all α- and β-subunits with the exception of β7. Deletion of the *UMP1* gene produces cells impaired in proteasomal activity with strong defects in autocatalytic processing of the active β-subunits [5]. Human *UMP1/POMP* is essential for cell viability, and several studies have revealed that point mutations in the coding region or flanking regions are involved in distinct diseases [7,8,9,10,11,12,13]. On the other hand, the *UMP1* gene has been considered as a drug target to overcome tumor cell resistance to proteasome inhibitors [14]. Structural studies of recombinant yeast Ump1 protein showed that it behaves similar to a natively unfolded protein with little secondary structure elements [15,16]. Structural studies based on electron microscopy (EM) and crosslinking revealed that, in the 15S PC, Ump1 loops around the inner cavity in a stretched-out conformation at the interface between α- and β-rings [6,17]. Experiments that either used partial trypsin cleavage [5] or detection of an N-terminal 6His tag on Ump1 by Ni-NTA nanogold particles in electron microscopy [6] indicated that the Ump1 N terminus projects out of the β-ring opening [6]. In higher-resolution EM structures, the N-terminal residues 1–26 were not resolved, suggesting that this part of the chaperone is also flexible in the 15S PC [17]. The first 61 residues of human Ump1 were shown to be dispensable for its incorporation into proteasome assembly intermediates [18]. Previous results showed that addition of either a small or a large tag to the N terminus provokes 15S PC accumulation, indicating that the N terminus has a specific role in proteasome biogenesis, likely in the dimerization of two 15S PC, a late step in the biogenesis pathway [6]. A bulkier extension such as a GFP module hocked to the N terminus of Ump1 blocks dimerization so dramatically that the pathway fails to produce fully assembled proteasomes, resulting in the severe inhibition of cell growth [6]. Interestingly, yeast cells are able to form proteasomes in the total absence of the chaperone Ump1, although the growth defects and proteasome impairment indicate that PC biogenesis is inefficient and error-prone [5].

In order to study the role of the Ump1 N terminus in proteasome assembly, we deleted its 16 N-terminal residues (^17–148^Ump1). Cells with the *^17–148^ump1* mutation display growth defects and a massive accumulation of 15S PCs as well as additional intermediates bearing the truncated Ump1. Interestingly, ^17–148^Ump1 acts as a dominant negative as its overexpression in the presence of wild-type endogenous Ump1 was synthetically lethal with the deletion of *RPN4.* The latter encodes a transcription factor that promotes the expression of proteasome genes, in particular when proteasome function is limiting or impaired [19,20]. Overexpression of the β7 subunit suppressed the growth defects of the *^17–148^ump1* mutant, suggesting that the N-terminal truncation of Ump1 compromises the incorporation of the β7 subunit. In vitro experiments showed that the propeptide of β7 promotes binding to the N-terminal domain of Ump1. Our findings reveal an important function of the Ump1 N terminus and the β7 propeptide in promoting the incorporation of β7 into the 15S PCs, a step that drives the dimerization of 15S PCs and the formation of 20S CP.

## 2. Materials and Methods

### 2.1. Yeast Methods

Yeast-rich (YPD) and synthetic (SD) minimal media with 2% dextrose or 2% galactose were prepared as described [21]. All *Saccharomyces cerevisiae* strains used are described in Appendix A. Truncated *UMP1* versions were produced by PCR with appropriated oligos. The truncated ORFs were followed by a sequence encoding a 2xHA tag and cloned in a centromeric (*CEN*) plasmid between the P*_UMP1_* promotor and the T*_CYC1_* terminator. All plasmids used to transform the yeast are listed in Appendix A. For overexpression of *PRE4*, *SCL1*, or *PRE2* from the copper-inducible P*_CUP1_*, a set of *2μ/LEU2*-based plasmids was generated. To compare growth rates, spot assays were performed. Cells were grown in liquid cultures to exponential phase, and diluted with sterile medium to OD_600_ = 1.0 in a total volume of 100 μL. From this suspension, sequential 1:10 dilutions were made in the same volume using sterile 96-well plates. Droplets of these cell suspensions were spotted onto agar plates with the relevant media using a frogger. Plates were incubated at different temperatures and for 1–3 days.

### 2.2. Preparation and Analysis of Yeast Protein Extracts

For native gel electrophoresis, native total protein crude extracts were prepared with cells from 15 mL cultures with OD_600_ ~1.0 with the help of glass beads in 26S buffer (50 mM Tris-HCl pH 7.5, containing 1 mM DTT, 5 mM MgCl_2_, 2 mM ATP, 15% (*v*/*v*) glycerol). Protein concentration was quantified with the Bio-Rad Bradford assay, and 10 µg of proteins were applied per lane. For gel filtration analyses, proteins were extracted in 26S buffer by grinding in liquid nitrogen in a pre-cooled mortar with a pestle [5,22]. Gel filtration on a Superose 6 column coupled to an ÄKTA FPLC (GE Healthcare) as well as SDS-PAGE, and immunoblots were performed as described previously [5,6], except that fraction sizes were 500 μL. Analysis by native polyacrylamide gel electrophoresis (PAGE) were performed in Tris-HCl 4–15% gradient gels (Bio-Rad) or as described [5]. Samples were mixed with NB4x (240 mM Tris-HCl pH 8.8, 80% (*v*/*v*) glycerol, 0.04% (*w*/*v*) bromophenol blue) and loaded onto the gel. The gels were incubated 10 min in transfer buffer containing 2% SDS before electro-blotting for 2 h at 0.8 mA/cm^2^. Proteasomal peptidase activity assays were performed as follows: glass bead extracts were used to determine chymotryptic activity in extracts from cells of three independent cultures. 10 µg of extract proteins were mixed with 5 µg of the substrate Succinyl-Leu-Leu-Val-Tyr-7-amido-4-methyl-coumarin (Bachem). The assay conditions were described previously [5,22].

### 2.3. Production and Purification of Proteins in Escherichia coli

*E. coli* strains transformed with expression plasmids (listed in Appendix A) encoding tagged proteins were grown in Luria-Bertani (LB) media (0.5% yeast extract, 1% tryptone, 1% sodium chloride) containing 100 µg/mL ampicillin at 37 °C until the exponential growth phase was reached. Afterwards, protein expression was induced with 1 mM IPTG, and cells were grown over night at 18 °C. Cells were harvested by 3000×*g* rcf for 10 min. Cell pellets were re-suspended in 15 µL cold lysis buffer (50 mM Tris-HCl pH 7.4, 150 mM NaCl, 5 mM MgCl_2_, 15% glycerol) containing 0.1% Triton X-100, 1× Complete EDTA-free protease inhibitor cocktail, 10 µg/mL DNaseI, 1 mg/mL lysozyme per OD_600_ unit and suspension was incubated on ice for 30 min. Cell lysis was performed using glass beads (Ø 0.10–0.11 mm) and a vortex mixer at 4 °C for 5 min. Cell debris was pelleted by 14,000×*g* rcf for 20 min at 4 °C. Supernatants containing the protein extracts were transferred to fresh tubes and used for further analyses. The preparation of full-length untagged proteasomal β subunits was performed as follows: pellets of *E. coli* cells expressing 8His-SUMO1 fused to β subunits were resuspended in lysis buffer and incubated on ice as described for native protein extracts. Afterwards, the suspension was sonicated for 5 min, and cell debris was pelleted by 14,000×*g* rcf for 45 min at 4 °C. The supernatant was incubated with Ni-NTA Superflow resin (QIAGEN, Hilden, Germany) equilibrated in lysis buffer containing 0.1% Triton X-100 and 20 mM imidazole for 1–2 h at 4 °C with agitation. The resin was washed 4 times with 10 column bed volumes lysis buffer with 20 mM imidazole. Bound proteins were eluted in four steps with each 5 mL lysis buffer supplemented with 250 mM imidazole for 20 min with rotation. Eluted fractions were dialyzed overnight in lysis buffer without glycerol, but containing 20 mM imidazole. Simultaneously, the 8His-SUMO1 tag was cleaved using 6His tagged SENP1 enzyme to obtain the pure protein [23]. The next day, non-specifically bound material and 6His-SENP1 enzyme were removed by a 2 h incubation with Ni-NTA resin followed by incubation with Talon beads (Clontech, Palo Alto, CA, USA). No washing or elution steps were necessary at this point, unbound proteins simply needed to be collected as the resin flow-through after binding. Afterwards, proteins were concentrated using Vivaspin Turbo 4 (10,000 MWCO, Sartorius, Göttingen, Germany), aliquoted and snap-frozen in liquid nitrogen prior to storage at −80 °C. Ump1-6His proteins were extracted and purified as described above. After elution (2 × 1 mL lysis buffer containing 250 mM imidazole), Ump1 proteins were concentrated using Vivaspin centrifugal concentrators (3000 MWCO).

### 2.4. In Vitro Binding Assay Using Ni-NTA Pulldown 

Ni-NTA Superflow resin (QIAGEN) was equilibrated 3× with 20 column volumes (CV) lysis buffer. Pre-purified proteins or native extracts of *E. coli* cells transformed either with a plasmid expressing Ump1-6His or an empty control vector were added to the respective tubes in a total volume of 20 CV, and binding was allowed for 1 h at 4 °C, rotating. Afterwards, unspecific proteins were removed 4× using lysis buffer containing 20 mM imidazole. Purified authentic β7 or β1 test proteins were added in a total amount of 20 CV and binding was again allowed for 1 h at 4 °C with rotation. Washing was performed as described above, and samples were transferred to fresh tubes. Proteins were eluted in buffer with 250–500 mM imidazole for 1 h at 4 °C with shaking. Samples were analyzed by 12% SDS-PAGE and Western blotting. Antibodies used are listed in Appendix A. Membranes were scanned using a LI-COR Odyssey Imager and the signals analyzed with Image Studio Lite Ver 5.2 (LI-COR, Lincoln, NE, USA).

### 2.5. Fluorescence Microscopy-Based On-Bead Binding Assay 

Pre-equilibrated Ni-NTA Superflow resin (QIAGEN) was provided in a tube. Ump1-6His-containing or control protein extracts (EV) were added to the respective tubes and binding was allowed for 1 h at 4 °C, rotating. Afterwards, unspecific proteins were removed 4× using lysis buffer with 20 mM imidazole. Protein amounts of different mNeonGreen (NG) fusion protein [24] extracts were determined and adjusted by measuring the fluorescence using the Tecan Infinite F200 pro (filter 485 nm, 535 nm) prior to addition to the binding reaction. Binding was again allowed for 1 h at 4 °C with rotation. Proper washing was performed as described above. Samples were analyzed by fluorescence microscopy, which was carried out with the fluorescence microscope Zeiss Axioplan 2 and with the software AxioVision Rel 4.7 (Zeiss, Jena, Germany). For microscopy, a magnification of 10× was used, provided by the objective Zeiss Plan-Neofluar 10x/0.30 Ph1 (DIC I). For GFP, the following filter was used: F41-054 HQ-Cy2 HQ480/40 (excitation) Q505LP HQ527/30 (emission). Exposure times for image taking were 12 ms for brightfield and 1.5 s for GFP. For quantitative evaluation, images were analyzed using Fiji (ImageJ v1.53f, https://ij.imjoy.io).

## 3. Results

### 3.1. N-Terminally Truncated Ump1 Incorporates into Proteasome Assembly Intermediates

The purpose of the current study was to identify functional domains of the Ump1 proteasome assembly chaperone and the interactions they engage in. Specifically, we focused on the role of the Ump1 N-terminal domain. We produced centromeric (CEN) plasmids expressing either full-length Ump1 or its N-terminally truncated versions (^17–148^Ump1, ^56–148^Ump1, and ^82–148^Ump1). All of these versions carried a 2xHA tag at their C termini. Cells expressing the truncated Ump1 versions displayed strong growth defects similar to *ump1*∆ cells (Figure 1A), and showed reduced chymotryptic activities (Figure 1B). To learn more about how the truncations affect the function of Ump1, we asked if the different versions would be competent to form 15S PCs. We prepared native protein extracts from cells expressing either the wild-type or the truncated versions of Ump1-2HA. Identical amounts of protein were separated by native PAGE, and Ump1-containing complexes were analyzed by anti-HA Western blotting (Figure 1C). Full-length Ump1 was detected as a single band representing the 15S PC. Notably, all of the tested truncated versions ^17–148^Ump1, ^56–148^Ump1, and ^82–148^Ump1 were detected in 15S PCs as well, indicating that residues 82–148 are sufficient for the incorporation of Ump1 into these complexes. Strikingly, however, 15S PCs comprising the truncated Ump1 versions were much more abundant than the corresponding complex in cells with full-length Ump1. 

Moreover, two additional bands, one migrating faster than the 15S PC and the other one slower were detected in cells expressing truncated Ump1 forms (indicated by arrowheads in Figure 1C). The faster migrating band corresponds to an intermediate observed previously, which was found to accumulate under conditions where the Pba1-Pba2 chaperone becomes limiting [6]. We found that the intermediate with the lower mobility depended on Blm10, since it was not detected in *blm10*∆ cells (Appendix A). When we compared the distribution of complexes containing either full-length Ump1 or ^17–148^Ump1 by gel filtration, we observed that complexes with truncated Ump1 were much more abundant and spanned a wider range of fractions (22–29) than the full-length Ump1 (mainly in fractions 27 and 28) (Figure 1D). Of note, we did not detect any signal corresponding to free Ump1 suggesting that the truncated Ump1 is incorporated efficiently into proteasome precursor complexes. From these observations, we conclude that the residues between 1 and 82 are not essential for the incorporation of Ump1 into 15S PCS or earlier intermediates, but they appear to be important for downstream steps leading to the dimerization of 15S PCs and ultimately the degradation of Ump1 [5]. Supporting the relevance of the N-terminal domain of Ump1 in the latter steps, we also found that point mutations (I3T, S11P, or L29S) in this domain resulted in a similar accumulation of the above-mentioned complexes as ^17–148^Ump1 (Appendix A).

### 3.2. Autocatalytic Processing of β5 Is Incomplete in ^17–148^Ump1 Cells

In order to further pin down the function of the Ump1 N terminus, we tested if downstream maturation processes would be affected by analyzing the processing of the catalytic β subunits. We had already observed that ^17–148^Ump1 cells have decreased chymotryptic activity; therefore, we asked whether β5/Pre2 subunits display processing defects. We followed this subunit, tagged with HA, by gel filtration in cells expressing either wild-type Ump1 or ^17–148^Ump1 (Appendix A). Similarly, to what was observed earlier for *ump1*∆ cells [5], β5/Pre2 was incompletely processed in ^17–148^Ump1 cells (Appendix A). By contrast, no apparent deficiency in β1/Pre3 processing was observed (Appendix A). To summarize the data shown so far, cells expressing ^17–148^Ump1 accumulate unusual amounts of 15S PC (as well as other intermediates) and display proteasomes with properly processed β1 but incompletely processed β5 subunits.

### 3.3. RPN4 Is Required to Tolerate N-Terminally Truncated Ump1

The most striking observation from our analyses was the massive accumulation of 15S PCs and other precursors in ^17–148^Ump1 cells (Figure 1C,D and Appendix A). We asked if the abnormal increase in precursors would be due to an activation of an Rpn4 response. As cells lacking Ump1 do not tolerate *RPN4* gene deletion [20], we transformed *rpn4*∆ cells expressing endogenous *UMP1* with 2-micron plasmids either encoding full-length Ump1 or a ^17–148^Ump1 version under control of the inducible P*_GAL1_* promoter. Whereas induced overexpression of full-length Ump1 had no detectable effect on growth of wild-type on galactose media, induction of ^17–148^Ump1 inhibited growth of wild-type, and completely abolished growth of *rpn4*∆ cells on these media (Figure 2A). These observations indicate that ^17–148^Ump1 is capable of competing with the endogenous full-length Ump1, and that Rpn4-driven upregulation of proteasome genes is required for the cells to tolerate the deleterious effects of ^17–148^Ump1 incorporation. To obtain additional evidence in support of this conclusion, we tested if higher Rpn4 activity would rescue the growth phenotype. Indeed, the growth defect caused by ^17–148^Ump1 was partially rescued by the expression of a stable version of Rpn4 (^∆1–10/∆211–229^Rpn4 alias Rpn4*) (Figure 2A) [25]. This led us to assume that the insufficiency of the ^17–148^Ump1 cells may in part be related to a certain proteasome component, the amounts of which become limiting, thus causing the accumulation of 15S PC and other intermediates (see below). 

### 3.4. ^17–148^Ump1 Phenotypes Are Suppressed by Increased Levels of the β7 Proteasome Subunit 

In order to further understand which component might be a determining factor in suppressing ^17–148^Ump1 upon Rpn4* expression, we tested several proteasome subunits. We had previously observed that β7/Pre4 availability is a rate-limiting factor for 20S CP formation in a wild-type strain [26]. We tested if overexpressing *SCL1*, *PRE2*, or *PRE4* genes (encoding α1, β5, or β7, respectively) in ^17–148^Ump1 cells could suppress the slow growth phenotype. Whereas α1 and β5 overexpression did not cause visible effects, high β7 levels strongly suppressed the ^17–148^Ump1 growth defect (Figure 2B and Appendix A). We tested if there would be a correlation between growth defect suppression by β7 and the recovery of chymotryptic proteasome activity. Although chymotryptic activity in ^17–148^Ump1 cells increased slightly when β7 was overexpressed, it only reached about 30% of the level observed in wild-type cells (Appendix A). In line with this, the processing of β5 was far from complete in cells overexpressing β7 (Appendix A). Native PAGE analysis of proteasome precursor complexes bearing ^17–148^Ump1-HA-tagged, on the other hand, revealed that the accumulation of 15S PCs was reduced upon overexpression of β7 (Figure 2C). These findings suggested that the growth improvement of ^17–148^Ump1 cells by increased cellular levels of β7 was due to a more efficient formation of proteasomes from the stalled 15S PCs. Additionally, we found that *rpn4*∆ cells, unable to upregulate proteasome subunits and thus sensitive to strong expression of ^17–148^Ump1 (see Figure 2A), were also rescued by β7 overexpression (Appendix A).

### 3.5. Free β7 Subunits Accumulate in ump1∆ and ^17–148^ump1 Cells

The C-terminal tail of β7/Pre4, which extends from one half of the proteasome to the other, facilitates the formation of 20S core particles from two 15S PCs [27,28,29]. We and others have previously purified and characterized 15S PCs from yeast cells. The only subunit that was consistently missing from the 15S PCs in all studies was β7 [6,26,29,30,31,32] (see also Figure 2C). Additional data suggested that β7 promotes dimerization of 15S PCs and thereby 20S CP formation [26,30,33]. Because cells expressing ^17–148^Ump1 accumulate 15S PCs, we asked whether β7 is incorporated into this intermediate. We performed gel filtration analyses with extracts from cells expressing HA-tagged β7/Pre4. Not only in the wild-type, but also in *ump1*∆ and *^17–148^ump1* cells, β7 was absent from the 15S PCs (Figure 2D) likely because its stable incorporation requires dimerization of 15S PCs and subsequent maturation of nascent 20S CPs. Remarkably, in contrast to the situation in cells with wild-type Ump1, where no free β7 subunit was detectable, an enormous accumulation of free β7 was observed in fractions 35–37 of *ump1*∆ and *^17–148^ump1* cells (Figure 2D). The accumulation of free β7 suggested that the N-terminal domain of Ump1 residues might be involved in the binding of this subunit and thus important for its incorporation into 15S PCs leading to their dimerization. We reported previously that a C-terminal extension (CTE) of β7/Pre4 encompassing the last 19 residues is also important for 15S PCs dimerization. This tail extends from one half of the proteasome to the other, ending between the subunits β1/Pre3 and β2/Pup1 [27]. Therefore, we asked how a deletion of the β7 CTE (∆CTE) affects cells with N-terminally truncated Ump1. Inducible (P*_GAL1_*-driven) expression of ^17–148^Ump1 in a *ump1*∆ *pre4*-∆*CTE* background was not tolerated by the cells (Appendix A). Together, these data indicated that the CTE of β7 and an N-terminal domain of Ump1 are required to promote the incorporation of β7 and 15S PC dimerization.

### 3.6. β7 Binds to Ump1 In Vitro

The observation that an intact N-terminal part of Ump1 is required for normal incorporation of β7 during the assembly process suggested that the two proteins might directly interact in the process. Therefore, we asked whether Ump1 and β7, both produced in *E.*
*coli*, bind to each other in isolation in vitro. We performed binding assays by immobilizing either full-length Ump1, or an N-terminal (1–81) or a C-terminal part (82–148) of it (Figure 3A), via C-terminal 6His tags on Ni-NTA resin, and incubation with purified recombinant full-length β7 subunit. All three versions of resin-bound Ump1 specifically retained β7, whereas resin treated with a mock (empty vector) extract displayed only low (background) levels of β7 binding (Figure 3B). The specificity of this interaction is further supported by a lack of binding of β1/PRE3 to Ump1 (Figure 3C). The N-terminal fragment of Ump1 (residues 1–81) yielded a stronger binding of β7 than the C-terminal part (residues 81–148) (Figure 3B). Binding of the N-terminal half to β7 was abrogated by two-point mutations (I3T or S11P) (Figure 3B) very close to the N terminus that together were shown before to interfere with 15S PC dimerization (Appendix A). We conclude that an N-terminal domain of Ump1 binds relatively strongly to β7, but the C-terminal half of Ump1 also contributes to in vitro binding to β7.

### 3.7. β7 Propeptide Promotes Binding to Ump1 In Vitro

To further dissect the interaction between β7 and Ump1, we asked whether the propeptide or the CTE of β7 would be relevant. The different β7 variants shown in Figure 4A were produced in *E. coli* and assayed for binding to full-length Ump1-6His. Deletion of the β7 propeptide (∆LS-β7) led to a strong reduction in binding in comparison with full-length β7 (Figure 4B,C). Deletion of the CTE (β7-∆CTE), by contrast, had no significant effect on binding to Ump1. This observation was consistent with a lack of binding in an assay in which we tested whether the CTE fused to the C terminus of maltose binding protein (MBP) would bind to Ump1 (Appendix A). We conclude that presence of the propeptide is important for the interaction of β7 with Ump1, whereas the CTE is not important for the interaction of these two polypeptides, at least in vitro. 

We next asked whether the propeptide-dependent interaction of β7 involved the above-mentioned N-terminal domain of Ump1. To address this question, we shifted to a modified version of the binding assay. Full-length Ump1-6His or ^17–148^Ump1-6His were bound to Ni-NTA beads and incubated either with full-length β7 (^1–266^β7), β7 lacking its propeptide (^34–266^β7), or just the β7 propeptide (^1–33^β7), each fused to the fluorescent protein mNeonGreen (NG) (constructs depicted in Figure 5A). On-bead binding of the fluorescent proteins to Ump1 was analyzed by fluorescence microscopy (Figure 5B). Consistent with the earlier findings (Figure 3), we observed a strong reduction in binding efficiency (~50%) when the N-terminal 16 residues of Ump1 were deleted (Figure 5B,C). Additionally consistent with earlier data (Figure 4) was the strong reduction in binding upon deletion of the β7 propeptide to ~20% compared with full-length β7. This assay furthermore revealed that just the propeptide alone promoted binding of the fluorescent reporter protein to full-length Ump1. Strikingly, this binding was completely lost when Ump1 was N-terminally truncated. Together, these findings reveal critical functions of the Ump1 N-terminal domain and the β7 propeptide in the recruitment of β7 precursor during proteasome assembly, a step that drives dimerization of 15S PCs and formation of 20S CPs.

## 4. Discussion

### 4.1. Ump1 N-Terminal Domain Is Required for Efficient Dimerization of 15S PCs

In the report, we identified an important function of an N-terminal domain of yeast Ump1. This key assembly chaperone has been implicated in the proper assembly of proteasome precursor complexes (15S PCs), their dimerization, and subsequent proper execution of active site maturation by processing of β subunit propeptides [5,34]. Truncation analysis revealed that the C-terminal half (residues 82–148) of the protein appears to be sufficient for assembly of Ump1 into nascent 15S PCs, whereas N-terminal parts are required for efficient 15S PC dimerization (Figure 1C). Interestingly, deleting just the first 16 residues from the Ump1 N terminus (^17–148^Ump1) led to a more severe growth defect and a higher accumulation of 15S PCs than longer deletions (^56–148^Ump1 or ^82–148^Ump1). One possible explanation is that interactions of CP subunits within the first 16 residues are required to induce conformational changes in downstream parts of Ump1 that are important for interactions with other CP subunits. In this context, it is noteworthy that a deletion of the Pre2/β5 propeptide is lethal in the presence of Ump1 but tolerated in *ump1*∆ [5]. This observation hinted at functional interactions between this propeptide and Ump1 during the maturation process. Consistent with this notion is the observation that the processing of Pro-β5 is incomplete both in the *ump1*∆ strain [5] as well as in cells bearing the truncated ^17–148^Ump1 version (Appendix A). The truncated version of Ump1 is incorporated into nascent 15S PCs rather efficiently. This is not only indicated by their accumulation to high levels (Figure 1C,D), but also by the observation that ^17–148^Ump1, upon overexpression, can outcompete endogenous full-length Ump1, resulting in the inhibition of growth and lethality in *rpn4*∆ cells, which are unable to increase the transcription of proteasome genes (Figure 2A).

### 4.2. Ump1 N-Terminal Domain Interacts with Pro-β7 to Promote 15S PC Dimerization 

A candidate approach identified, among other tested CP subunits (Pre2/β5 and Scl1/α1), the Pre4/β7 subunit as a rather efficient suppressor of growth defects caused by ^17–148^Ump1 (Appendix A and Figure 2B). Previous work had shown that β7 is absent from the 15S PCs and drives their dimerization to form 20S CPs (see Introduction). A C-terminal extension (CTE) of this subunit that intercalates between the Pre3/β1 and Pup1/β2 of the opposing half after dimerization of 15S PCs was shown to be important in this process. In vitro binding experiments, however, indicated that the CTE is not engaging in an interaction with Ump1 (Figure 4). Instead, our experiments revealed that the propeptide of the β7 precursor subunit promotes binding to Ump1 in vitro, and this interaction depended on the presence of the N-terminal 16 residues of Ump1 (Figure 5). Although the structure of the N-terminal 26 residues of Ump1 could not be resolved [17], crosslinking of its residue ^19^Lys to residue ^91^Lys of β6 [6] is compatible with our biochemical data as it suggests that the N-terminal domain of Ump1 is likely not too far away from the position where Pro-β7 will insert. The distance of ^91^Lys of β6 to the N-terminal Thr of processed β7 in the structure of the mature 20S CP is only ~15 Å [35], indicating that the N-terminal domain of Ump1 is probably well-positioned to interact with the propeptide of incoming Pro-β7 (Figure 6). 

Although the N-terminal domain of Ump1 and the propeptide of β7 appear to be mainly responsible for their interaction, other parts may additionally contribute, at least in the in vitro binding experiment (Figure 3 and Figure 5). Importantly, however, our data obtained with mNeonGreen reporter protein fused to the β7 propeptide clearly demonstrated its capacity to bind to Ump1 but only if the N-terminal 16 residues of Ump1 were present (Figure 5). Together, these in vitro studies that employed *E. coli*-produced polypeptides, and the in vivo experiments with yeast mutants, identified a novel function of the N-terminal part of Ump1, the recruitment of Pro-β7 to the 15S PC complexes to drive their dimerization. The high sensitivity of our in vitro binding assays was critical to track down this interaction, because Pro-β7 cannot be detected in 15S PC isolated from yeast even when a truncation of the CTE reduces efficiency of 15S PC dimerization [26]. The latter observation suggested that stable incorporation of the β7 subunit requires multiple interactions and events. Based upon our new results, we propose that an interaction between the N-terminal part of Ump1, which likely protrudes from the β ring (see Introduction; [6]), and the propeptide of β7 initially provides one such interaction. Another interaction is the above-mentioned intercalation of the β7 CTE between the β1 and β2 subunits of the juxtaposed 15S PC upon their dimerization. The full stabilization of a nascent 20S CP pre-holo enzyme is probably only achieved when a 15S PCs dimer is clamped together by two β7 subunits.

### 4.3. Possible Additional Functions of the Ump1 N-Terminal Domain

It is likely that the N-terminal domain of Ump1 is involved in additional interactions, possibly later in the maturation process. The reason for this assumption is that the phenotypes caused by a 16 residue N-terminal truncation of Ump1 (Figure 1A) are more severe than those caused by point mutations in this segment (Appendix A), which cause a similar reduction in β7 binding as the truncation (Figure 3). Again, an interaction with the β5 propeptide may be one such additional function, which could be required to position this part of the subunit such that autocatalytic processing can occur efficiently [37]. An impairment of Pro-β5 processing in cells with ^17–148^Ump1 is consistent with this possibility. In this context it is also noteworthy that the phenotype caused by a deletion of the β5 propeptide was shown to be suppressed by overexpression of β7 [30], similar to what we observed with ^17–148^Ump1. One possible explanation suggested by our findings is that the β5 propeptide is important for a proper exposure of the Ump1 N terminus, or alternatively, may operate together with it, to enable efficient recruitment of Pro-β7. This function of Ump1 in the control of 15S PC dimerization depending on the β5 propeptide has been interpreted as a checkpoint, that prevents dimerization of incomplete or improper 15S PCs [30]. The availability of point mutations affecting one or the other function of the N-terminal domain of Ump1 may help to further dissect such functions in future studies.

## Figures and Tables

**Figure 1 biomolecules-12-00253-f001:**
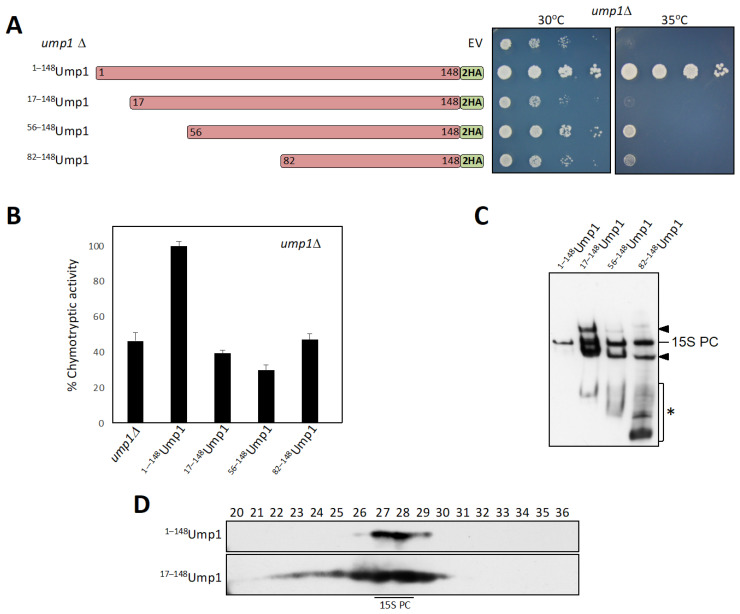
N-terminally truncation of Ump1 leads to functional impairment. (**A**) Yeast *ump1*∆ cells expressing N-terminally truncated Ump1 display poor and temperature-sensitive growth. On the left, HA-tagged version of Ump1 encoded by low copy (*CEN*) plasmids under control of the P*_UMP1_* promoter are depicted. *Ump1*∆ cells were transformed either with *CEN* plasmids encoding the depicted Ump1 variants or with the empty vector (EV). Exponentially growing cultures of the transformants were serially (1:10 steps) diluted, spotted onto selective minimal (SD) media, and incubated for 2 days at the indicated temperatures (right part). (**B**) N-terminal truncation of Ump1 leads to reduced proteasomal chymotryptic (CT) activity. Exponentially growing cells expressing the indicated Ump1 versions (same as in (**A**)) were harvested in triplicates and the CT activity was measured in native extracts. Depicted are the mean activities with standard deviations. The mean of the values for the transformants with full-length Ump1 was set to 100%. (**C**) Native PAGE analysis of complexes containing Ump1 variants. The same extracts as used in (**B**) were separated by native PAGE and analyzed by anti-HA Western blotting. The position of the 15S PC is indicated. Arrowheads, additional complexes described in the main text. *, lower molecular weight forms containing truncated Ump1 variants. Ponceau S staining of the membrane as a loading control is provided in Appendix A. (**D**) Gel filtration analysis of complexes present in extracts from cells expressing either full-length Ump1-HA or truncated ^17–148^Ump1-HA. Protein extracts were separated on a Superose 6 column, and 0.5 mL fractions were collected. Chromatograms are provided in Appendix A. Fractions from 20 to 36 were analyzed by SDS-PAGE and anti-HA Western blotting.

**Figure 2 biomolecules-12-00253-f002:**
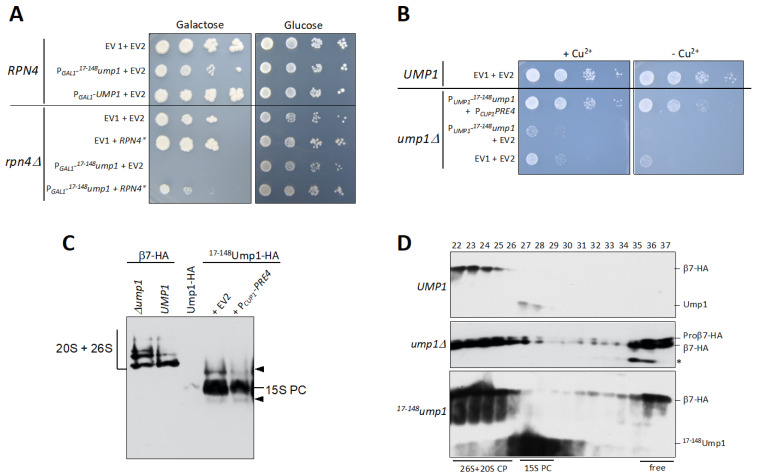
Toxicity of ^17–148^Ump1 is suppressed by overexpression of *PRE4*/β7. (**A**) Cells lacking transcription factor Rpn4 are hypersensitive to ^17–148^Ump1. Wild-type (*RPN4*) or *rpn4*∆ cells were transformed either with an empty vector (EV1) or with plasmids encoding either full-length Ump1 or ^17–148^Ump1, synthesis of which was induced by induction of the P*_GAL1_* promotor on galactose media. At the same time, the cells were co-transformed either with another empty vector (EV2) or with a plasmid expressing a stable version of Rpn4* [25]. Note that *rpn4*∆ cells constitutively express endogenous Ump1. *Ump1*∆ could not be used here because it causes synthetic lethality together with *rpn4*∆ [20]. Serial dilutions of cells were spotted on plates either containing galactose or glucose as carbon source and incubated at 30 °C for 3 days or 2 days, respectively. (**B**) Overexpression of *PRE4* encoding the β7 subunit suppresses growth defect from cells expressing ^17–148^Ump1. Wild-type (*UMP1*) or *ump1*∆ cells were either transformed with an empty vector (EV1) or with a plasmid encoding ^17–148^Ump1 expressed from the native P*_UMP1_* promoter. At the same time, the cells were co-transformed either with another empty vector (EV2) or with a high-copy (2µ-based) plasmid expressing *PRE4* from the copper-inducible P*_CUP1_* promoter. Serial dilutions of cells were spotted on selective glucose plates either with or without 100 µM CuSO_4_, and incubated at 30 °C for 2 days. (**C**) Overexpression of full-length β7 reduces the amount of accumulated 15S PC containing ^17–148^Ump1. Shown is a native gel analysis of proteasome complexes. Left part, *ump1*∆ or wild-type (*UMP1*) cells with β7-HA. Right part, ^17–148^Ump1-HA expressing cells transformed either with an empty vector or with a plasmid overexpressing *PRE4* (same as in (**B**)). Complexes with HA-tagged proteins were detected by anti-HA Western blotting. Positions of 15S PC as well as 20S and 26S particles are indicated. Arrowheads point to two additional complexes detected in extracts from^17–148^Ump1-expressing cells. Note that high levels of 15S PC accumulate in the latter cells in comparison with otherwise wild-type cells expressing Ump1-HA (middle lane). (**D**) Free β7 subunit accumulates in *ump1*∆ or *^17–148^ump1* cells. Gel filtration analysis of complexes present in cells expressing either full-length Ump1-HA, no Ump1 (*ump1*∆) or ^17–148^Ump1-HA as well as β7-HA. Fractions from 22 to 37 were analyzed by SDS-PAGE and anti-HA Western blotting. *, degradation product.

**Figure 3 biomolecules-12-00253-f003:**
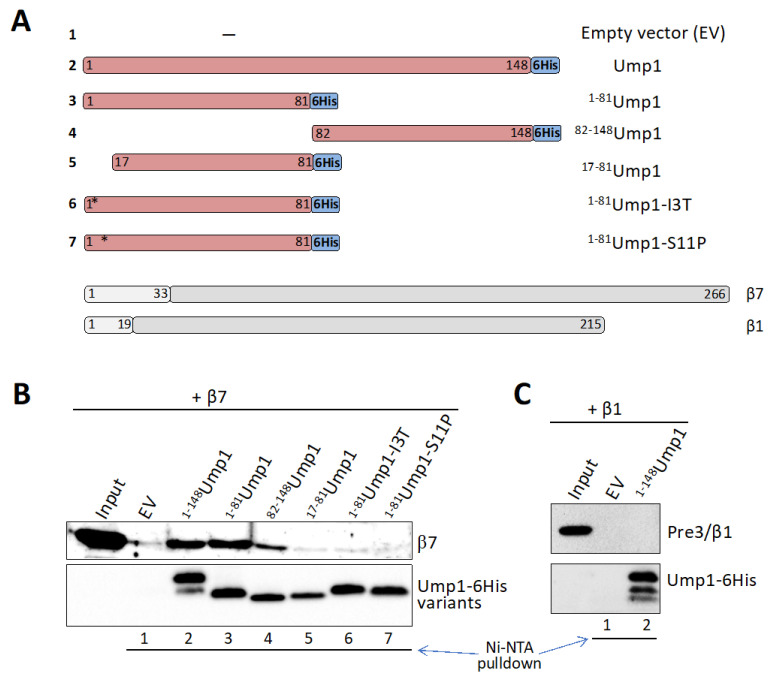
Ump1 N-terminal domain interacts with β7 subunit in vitro. (**A**) Schematic representation of constructs expressed in *E. coli.* Ump1 polypeptides were C-terminally tagged with 6His. β1 and β7 were initially expressed as fusions to 8His-SUMO1 (not shown) and later, after cleaving off SUMO1, obtained in an untagged form. Positions of the point mutations I3T and S11P are indicated by stars. (**B**) Interaction of the different Ump1 polypeptides with recombinant full-length β7. Ni-NTA beads were first loaded with Ump1 variants purified from *E. coli* extracts. A mock purification from an empty vector (EV) transformant extract served as a control. Loaded beads were then incubated with full-length β7. After washing, bound proteins were eluted with imidazole and analyzed by SDS-PAGE and anti-β7 and anti-6His Western blotting. The numbers at the bottom refer to the constructs represented in (**A**). (**C**) Interaction assay as in (**B**) but between β1 and full-length Ump1.

**Figure 4 biomolecules-12-00253-f004:**
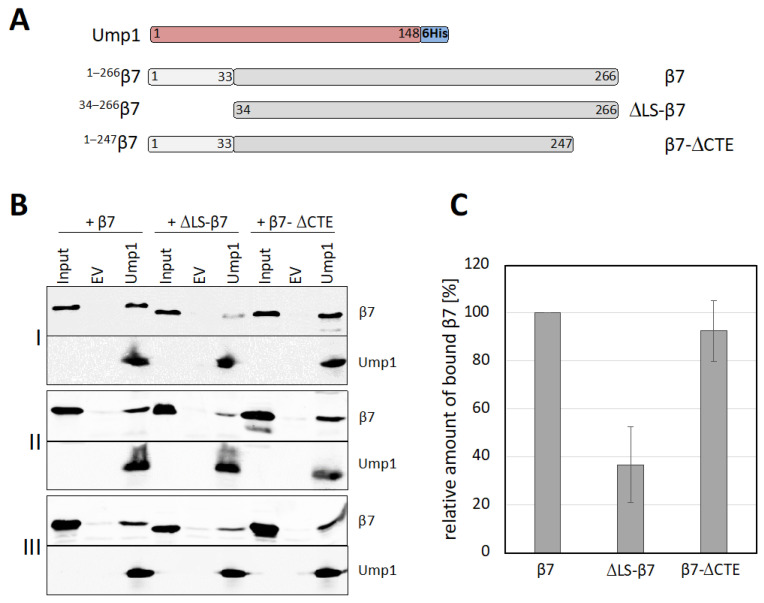
The propeptide of the β7 precursor polypeptide promotes binding to Ump1. (**A**) Schematic representation of the constructs used in this experiment: Ump1 fused to 6His, and three distinct untagged β7 versions (full-length, without leader sequence (∆LS), without C-terminal extension (∆CTE)). (**B**) Native extracts from *E. coli* cells transformed either with an empty vector control (EV) or with a plasmid expressing Ump1-6His were first incubated with Ni-NTA resin, which was then washed and further incubated with purified β7 versions. Bound proteins were eluted with imidazole and analyzed by anti-β7 and anti-Ump1 Western blotting. Shown are the results of three independent sets of experiments (I, II, and III). (**C**) Comparison of the binding efficiencies of tested β7 variants and full-length Ump1. Quantitative evaluation of the signals shown in (**B**) was performed with the LI-COR infrared scanner. Signals of the β7 variants eluted from Ump1-loaded resin were first set in relation to the input, of which 10% was loaded on the gel. Background β7-EV signals were subtracted. The mean value for recovery of full-length β7 was set to 100%, and the signals for the truncated variants were related to it. Error bars represent standard deviation of the mean (*n* = 3).

**Figure 5 biomolecules-12-00253-f005:**
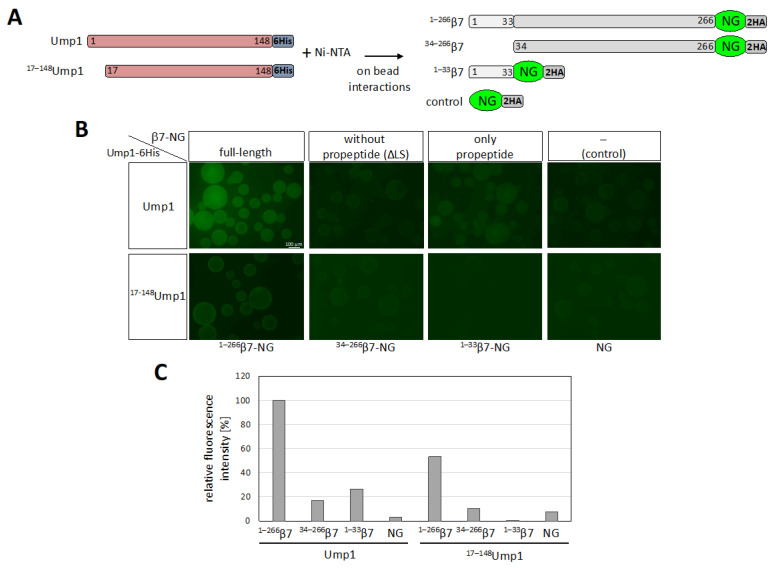
Interaction between Ump1 and β7 is mainly between the Ump1 N terminus and the β7 propeptide. (**A**) Schematic representation of full-length Ump1 and ^17–148^Ump1 fused to 6His, as well as of β7 variants C-terminally fused to mNeonGreen (NG) and 2HA, all produced in *E. coli*. NG-2HA alone was used as a control. (**B**) Fluorescence-microscopy-based on-bead binding assay. The different NG variants depicted in (**A**) were incubated with Ni-NTA beads loaded either with full-length or truncated Ump1 and imaged under the fluorescence microscope with identical exposure times. The scale bar indicates 100 µm. All images were adjusted with +40% brightness and −40% contrast using PowerPoint (Microsoft). (**C**) Quantitative analysis of the results shown in (**B**). Image quantification of signals detectable on beads of the same diameter was performed with Fiji (ImageJ) (*n* = 5). Background β7-NG signals obtained with Ni-NTA beads incubated with extract from an empty vector *E. coli* transformant not expressing Ump1 (not shown here) were subtracted (see Appendix A). Signals obtained for full-length β7-NG bound to full-length Ump1 were set to 100 % and the signals for the other NG variants were calculated relative to them. Image quantification was performed with Fiji (ImageJ).

**Figure 6 biomolecules-12-00253-f006:**
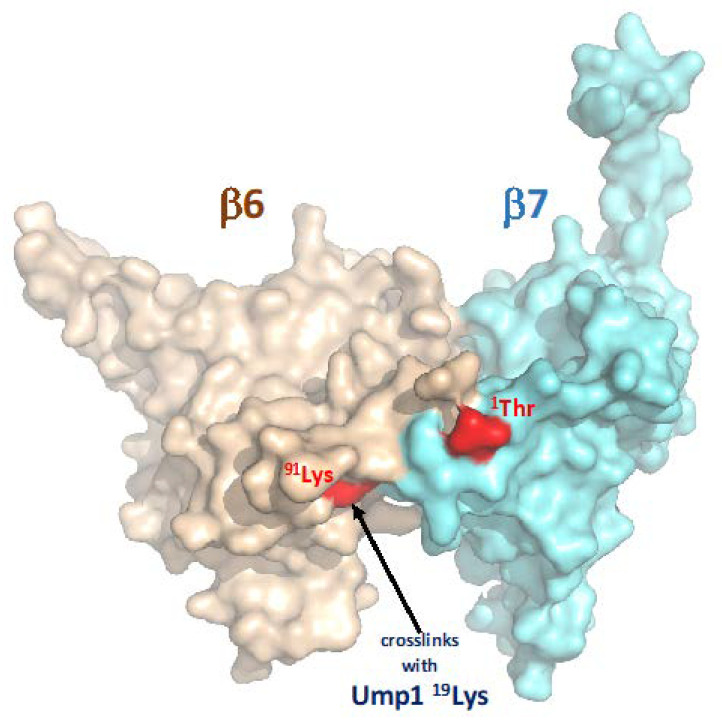
Proximity of Ump1 crosslinking site and the N terminus of β7. Surface representation of subunits β6 and β7 as found in the structure of the mature *S. cerevisiae* 20S CP (PDB code 1RYP) [35]. Highlighted in red are the N-terminal ^1^Thr of mature β7 and ^91^Lys of β6. The latter crosslinked with residue ^19^Lys of Ump1 in the 15S PC [6]. The figure was generated with PyMOL [36].

## Data Availability

Data is contained within the article or Appendix A.

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
