# Peer review of "Interaction with the Assembly Chaperone Ump1 Promotes Incorporation of the β7 Subunit into Half-Proteasome Precursor Complexes Driving Their Dimerization"

_biomolecules, 2022, doi:10.3390/biom12020253_

Round 1

Reviewer 1 Report

The Dohmen lab has made many important contributions to CP assembly, including the discovery of Ump1 (Ramos et al., 1998) and the role of Beta7's C-terminal extension in promoting half-CP fusion (Ramos et al., 2004). Here, they report that the N-terminus of Ump1 directly interacts with the N-terminal propeptide of Beta7. This is a very interesting interaction because it directly links Ump1 to half-CP fusion, which in turn drives the late steps of CP assembly.

            Overall, I think this is a fascinating contribution that will be of interest to people in the field. There are a few places where I am not sure the conclusions are justified by the data. I would not consider these points to be "mandatory," but I think they could strengthen the paper.

  1. The least convincing data in the paper are in Fig. 4B. Frankly, my visual inspection of the data does not match the quantitation in Figure 4C. I can barely see the reduction in binding of deltaLS-Beta7 to Ump1, especially in replicates II and III. I don't know if the difference can be brought out by optimizing the wash conditions (e.g. more salt or detergent)? I think the point is made pretty well in Fig. 3B (lanes 3 and 5) and Fig. 5B, so I am not concerned about the overall conclusion.

  1. I do not see much difference in growth between full length Ump1 (lane 2) and the two Ump1 point mutants (lanes 4, 5) in Figure S1B. I have personally done hundreds of these kinds of assays and I would not interpret those differences as meaningful. Perhaps the defect could be made more clear by using a lower-fold serial dilution (e.g. 3-fold instead of 10-fold) or a proteotoxic chemical (e.g. canavanine)? Another possibility would be to simply remove this panel from the paper since Fig. S1C makes the point in a more convincing manner. But if they can improve S1B, it would improve the paper.

  1. The authors say that the cross-linking data indicate that the N-terminal domain of Ump1 is "very close" to Beta7 (line 446). However, as I recall, the cross-linkable distance in the Kock et al., 2015 paper was 30 angstroms, which is a rather large distance in the context of the proteasome. Maybe saying that the cross-linking data are fully compatible with the biochemical data presented here would be a more accurate statement?

Reviewer 2 Report

Zimmermann et al. identified the N-terminus of Ump1 is interacting with the β7 propeptide, and demonstrated its importance in dimerization of 15S Precursor Complex during proteasome assembly, using in vitro biochemical and yeast genetic studies. The biochemical studies demonstrating the interactions are clear and convincible, as well as the yeast genetic studies. However, it’s not well established about the mechanism that over-expression of β7 would suppress the phenotype caused by 17-148 Ump1, considering no interactions of β7 and 17-148 Ump1.

I would like to accept this paper for publication, though further studies are required for interaction between β7 and Ump1. Moreover, the finding of association of β7 and N terminus of Ump1 would provide insights into the diverse functions of this flexible region of Ump1 in the field.

Some minor comments:

In Figure 1C, please include loading controls for the western blot.

Please include the chromatography for the gel filtrations throughout the manuscript.
